# Diffusion models as plug-and-play priors

**Alexandros Graikos**
Stony Brook University
Stony Brook, NY
agraikos@cs.stonybrook.edu

**Nikolay Malkin**
Mila, Université de Montréal
Montréal, QC, Canada
nikolay.malkin@mila.quebec

**Nebojsa Jojic**
Microsoft Research
Redmond, WA
jojic@microsoft.com

**Dimitris Samaras**
Stony Brook University
Stony Brook, NY
samaras@cs.stonybrook.edu

## Abstract

We consider the problem of inferring high-dimensional data $\mathbf{x}$ in a model that consists of a prior $p(\mathbf{x})$ and an auxiliary differentiable constraint $c(\mathbf{x}, \mathbf{y})$ on $\mathbf{x}$ given some additional information $\mathbf{y}$. In this paper, the prior is an independently trained denoising diffusion generative model. The auxiliary constraint is expected to have a differentiable form, but can come from diverse sources. The possibility of such inference turns diffusion models into plug-and-play modules, thereby allowing a range of potential applications in adapting models to new domains and tasks, such as conditional generation or image segmentation. The structure of diffusion models allows us to perform approximate inference by iterating differentiation through the fixed denoising network enriched with different amounts of noise at each step. Considering many noised versions of $\mathbf{x}$ in evaluation of its fitness is a novel search mechanism that may lead to new algorithms for solving combinatorial optimization problems. The code is available at https://github.com/AlexGraikos/diffusion_priors.

## 1   Introduction

Deep generative models, such as denoising diffusion probabilistic models [DDPMs; 39, 13] can capture the details of very complex distributions over high-dimensional continuous data $p(\mathbf{x})$ [30, 7, 1, 38, 43, 15]. The immense effective depth of DDPMs, sometimes with thousands of deep network evaluations in the generation process, is an apparent limitation on their use as off-the-shelf modules in hierarchical generative models, where models can be mixed and one model may serve as a prior for another conditional model. In this paper, we show that DDPMs trained on image data can be directly used as priors in systems that involve other differentiable constraints.

In our main problem setting, we assume that we have a prior $p(\mathbf{x})$ over high-dimensional data $\mathbf{x}$ and we wish to perform inference in a model that involves this prior and a constraint $c(\mathbf{x}, \mathbf{y})$ on $\mathbf{x}$ given some additional information $\mathbf{y}$. That is, we want to find an approximation to the posterior distribution $p(\mathbf{x}|\mathbf{y}) \propto p(\mathbf{x})c(\mathbf{x}, \mathbf{y})$. In this paper, $p(\mathbf{x} = \mathbf{x}_0, \mathbf{h} = \{\mathbf{x}_T, ..., \mathbf{x}_1\})$ is provided in the form of an independently trained DDPM over $\mathbf{x}_T, \ldots, \mathbf{x}_0$ (§2.2), making the DDPM a 'plug-and-play' prior.

Although the recent community interest in DDPMs has spurred progress in training algorithms and fast generation schedules [30, 37, 45], the possibility of their use as plug-and-play modules has not been explored. Furthermore, as opposed to existing work on plug-and-play models (starting from [29]), the algorithms we propose *do not require* additional training or finetuning of model components or inference networks.

36th Conference on Neural Information Processing Systems (NeurIPS 2022).

One obvious application of plug-and-play priors is conditional image generation (§3.1, §3.2). For example, a denoising diffusion model trained on MNIST digit images might define $p(\mathbf{x})$, while the constraint $c(\mathbf{x}, \mathbf{y})$ may be be the probability of digit class $\mathbf{y}$ under an off-the-shelf classifier. However, changing the semantics of $\mathbf{x}$, we can also use such models for inference tasks where neural networks struggle with domain adaptation, such as image segmentation: $c(\mathbf{x}, \mathbf{y})$ constrains the segmentation $\mathbf{x}$ to match an appearance or a weak labeling $\mathbf{y}$ (§4). Finally, we describe a path towards using DDPM priors to solve continuous relaxations of combinatorial search problems by treating $\mathbf{y}$ as a latent variable with combinatorial structure that is deterministically encoded in $\mathbf{x}$ (§5).

## 1.1 Related work

**Conditioning DDPMs.** DDPMs have previously been used for conditional generation and image segmentation [36, 42, 1]. With few exceptions – such as [3], which uses a pretrained DDPM as a feature extractor – these algorithms assume access to paired data and conditioning information during training of the DDPM model. In [7], a classifier $p(y \mid \mathbf{x}_t)$ that guides the denoising model towards the desired subset of images with the attribute $y$ is trained in parallel with the denoiser. In [5], generation is conditioned on an auxiliary image by guiding the denoising process through correction steps that match the low-frequency components of the generated and conditioning images. In contrast, we aim to build models that combine an independently trained DDPM with an auxiliary constraint.

Our approach is also related to work on adversarial examples. Adversarial samples are produced by optimizing an image $\mathbf{x}$ to satisfy a desired constraint $c$ – a classifier $p(\mathbf{y}|\mathbf{x})$ – without reference to the prior over data. As supervised learning algorithms can ignore the structure in data $\mathbf{x}$, focusing only on the conditional distribution, it is possible to optimize for input $\mathbf{x}$ that provides the desired classification in various surprising ways [41]. In [31], a diffusion model is used to defend from adversarial samples by making images more likely under a DDPM $p(\mathbf{x})$. We are instead interested in *inference*, where we seek samples $\mathbf{x}$ that satisfy *both* the classifier and the prior. (Our work may, however, have consequences for adversarial generation.)

**Conditional generation from unconditional models.** Works that preceded the recent popularity of DDPMs [29, 9] show how an unconditional generative model, such as a generative adversarial network [GAN; 11] or variational autoencoder [VAE; 21], can be combined with a constraint model to generate conditional samples. Regarding generative diffusion models, recent literature has focused on utilizing unconditional, pretrained DDPMs as priors to solve linear inverse imaging problems. Both in [40] and [20], the authors modify the DDPM sampling algorithm, with knowledge of the linear degradation operator, to reconstruct an image consistent with the learned prior and given measurements. A generalization of these methods in [18] shows how any pretrained denoising network can be used as the prior for solving linear inverse problems. We also clarify that although the term 'plug-and-play' is widely used in the inverse imaging literature we refer to it in the scope of in-domain generation under differentiable constraints, in the same sense as [29].

**Latent vectors in DDPMs.** Modeling the latent prior distribution in VAE-like models using a DDPM has been studied in [38, 43]. On the other hand, in §5, we perform inference in the low-dimensional *latent* space under a pretrained DDPM on a high-dimensional data space. Our approach to semantic segmentation (§4) is also related to [34], where a prior $p(\mathbf{z})$ over latents is used to tune a posterior network $q(\mathbf{z}|\mathbf{x})$. There, the priors are of relatively simple structure and are sample-specific, rather than global diffusion priors like in this paper.

## 2 Method

### 2.1 Problem setting

Recall that we want to find an approximation to the posterior distribution $p(\mathbf{x}|\mathbf{y}) \propto p(\mathbf{x})c(\mathbf{x}, \mathbf{y})$, where $p(\mathbf{x})$ is a fixed prior distribution. Fixing $\mathbf{y}$ and introducing an approximate variational posterior $q(\mathbf{x})$, the free energy

$$F = -\mathbb{E}_{q(\mathbf{x})}[\log p(\mathbf{x}) + \log c(\mathbf{x}, \mathbf{y}) - \log q(\mathbf{x})] \tag{1}$$

is minimized when $q(\mathbf{x})$ is closest to the true posterior, i.e., when $\mathrm{KL}(q(\mathbf{x})\|p(\mathbf{x}|\mathbf{y}))$ is minimized. When $q(\mathbf{x})$, and the learning algorithm used to fit it, are expressive enough to capture the true

posterior, this minimization yields the exact posterior $p(\mathbf{x}|\mathbf{y})$. Otherwise, $q$ will capture a 'mode-seeking' approximation to the true posterior [27]; in particular, if $q(\mathbf{y})$ is a Dirac delta, it is optimal to concentrate $q$ at the mode of $p(\mathbf{x}|\mathbf{y})$. When the prior involves latent variables $\mathbf{h}$ (i.e., $p(\mathbf{x}) = \int_{\mathbf{h}} p(\mathbf{x}|\mathbf{h})p(\mathbf{h})\,d\mathbf{h}$), the free energy is

$$
\begin{aligned}
F &= -\mathbb{E}_{q(\mathbf{x})q(\mathbf{h}|\mathbf{x})}[\log p(\mathbf{x},\mathbf{h}) + \log c(\mathbf{x},\mathbf{y}) - \log q(\mathbf{x})q(\mathbf{h}|\mathbf{x})] \\
&= -\mathbb{E}_{q(\mathbf{x})q(\mathbf{h}|\mathbf{x})}[\log p(\mathbf{x},\mathbf{h}) - \log q(\mathbf{x})q(\mathbf{h}|\mathbf{x})] - \mathbb{E}_{q(\mathbf{x})}[\log c(\mathbf{x},\mathbf{y})].
\end{aligned}
\tag{2}
$$

We are, in particular, interested in a general procedure for minimizing $F$ with respect to an approximate posterior $q(\mathbf{x})$ for any differentiable $c$ when $p$ is a DDPM (§2.2).

A free energy of the same structure was also studied in [43], where a DDPM $p(\mathbf{z})$ over a latent space is hybridized as a parent to a decoder $p(\mathbf{x}|\mathbf{z})$, with an additional inference model $q(\mathbf{z}|\mathbf{x})$ trained jointly with both of these models. On the other hand, we aim to work with independently trained components that operate directly in the pixel space, e.g., an off-the-shelf diffusion model $p(\mathbf{x})$ trained on images of faces and an off-the-shelf face classifier $p(\mathbf{y}|\mathbf{x})$, without training or finetuning them jointly (§3.2).

## 2.2 Denoising diffusion probabilistic models as priors

Denoising diffusion probabilistic models (DDPMs) [39, 13] generate samples $\mathbf{x}_0$ by reversing a (Gaussian) noising process. DDPMs are deep directed stochastic networks:

$$
p(\mathbf{x}_T, \mathbf{x}_{T-1}, ..., \mathbf{x}_0) = p(\mathbf{x}_T) \prod_{t=1}^{T} p_\theta(\mathbf{x}_{t-1} \mid \mathbf{x}_t),
\tag{3}
$$

$$
p_\theta(\mathbf{x}_{t-1} \mid \mathbf{x}_t) = \mathcal{N}(\mathbf{x}_{t-1}; \boldsymbol{\mu}_\theta(\mathbf{x}_t, t), \boldsymbol{\Sigma}_\theta(\mathbf{x}_t, t)), \qquad p(\mathbf{x}_T) = \mathcal{N}(\mathbf{0}, \mathbf{I}),
\tag{4}
$$

where $\boldsymbol{\mu}_\theta$ and $\boldsymbol{\Sigma}_\theta$ are neural networks with learned parameters (often, as in this paper, $\boldsymbol{\Sigma}_\theta$ is fixed to a scalar diagonal matrix depending on $t$). The model starts with a sample from a unit Gaussian $\mathbf{x}_T$ and successively transforms it with a nonlinear network $\boldsymbol{\mu}_\theta(\mathbf{x}_t, t)$ adding a small Gaussian innovation signal at each step according to a noise schedule. After $T$ steps, the sample $\mathbf{x} = \mathbf{x}_0$ is obtained.

In general, using such a model as a prior over $\mathbf{x}$ would require an intractable integration over latent variables $\mathbf{h} = (\mathbf{x}_T, ..., \mathbf{x}_1)$:

$$
p(\mathbf{x}) = \int_{\mathbf{h}} p(\mathbf{x}_T, \mathbf{x}_{T-1}, ..., \mathbf{x}_1, \mathbf{x}_0 = \mathbf{x})\,d\mathbf{x}_T \, ... \, d\mathbf{x}_1.
\tag{5}
$$

However, DDPMs are trained under the assumption that the posterior $q(\mathbf{x}_t|\mathbf{x}_{t-1})$ is a simple diffusion process that successively adds Gaussian noise according to a predefined schedule $\beta_t$:

$$
q(\mathbf{x}_t \mid \mathbf{x}_{t-1}) = \mathcal{N}(\mathbf{x}_t; \sqrt{1 - \beta_t}\mathbf{x}_{t-1}, \beta_t \mathbf{I}), \quad t = 1, \ldots, T.
\tag{6}
$$

Therefore, if $p(\mathbf{x})$ is the likelihood (5) of $\mathbf{x}$ under a DDPM, then in the first expectation of (2) we should use $q(\mathbf{h} = \{\mathbf{x}_T, ..., \mathbf{x}_1\}|\mathbf{x}_0 = \mathbf{x}) = \prod_{t=1}^{T} q(\mathbf{x}_t \mid \mathbf{x}_{t-1})$. The simplest approximation to the posterior over $\mathbf{x} = \mathbf{x}_0$ is a point estimate:

$$
q(\mathbf{x}) = \delta(\mathbf{x} - \boldsymbol{\eta})
\tag{7}
$$

where by $\delta$ we denote the Dirac delta function. Thus, we can sample $\mathbf{x}_t$ at any arbitrary time step using the forward noising process as

$$
q(\mathbf{x}_t) = \mathcal{N}(\mathbf{x}_t; \sqrt{\bar{\alpha}_t}\boldsymbol{\eta}, (1 - \bar{\alpha}_t)\mathbf{I})
\tag{8}
$$

where $\alpha_t = 1 - \beta_t$ and $\bar{\alpha}_t = \prod_{i=1}^{t} \alpha_t$. Analogously to [13], we can also extract a conditional Gaussian $q(\mathbf{x}_{t-1} \mid \mathbf{x}_t, \boldsymbol{\eta})$ and express the first expectation in (2) as

$$
-\mathbb{E}_{q(\mathbf{x})q(\mathbf{h}|\mathbf{x})}[\log p(\mathbf{x},\mathbf{h}) - \log q(\mathbf{x})q(\mathbf{h}|\mathbf{x})] = \sum_{t} \mathrm{KL}(q(\mathbf{x}_{t-1} \mid \mathbf{x}_t, \boldsymbol{\eta}) \,\|\, p_\theta(\mathbf{x}_{t-1} \mid \mathbf{x}_t)),
\tag{9}
$$

which after reparametrization [13] leads to

$$
\sum_{t} w_t(\beta)\mathbb{E}_{\boldsymbol{\epsilon} \sim \mathcal{N}(\mathbf{0}, \mathbf{I})}[\|\boldsymbol{\epsilon} - \boldsymbol{\epsilon}_\theta(\mathbf{x}_t, t)\|_2^2], \quad \mathbf{x}_t = \sqrt{\bar{\alpha}_t}\boldsymbol{\eta} + \sqrt{1 - \bar{\alpha}_t}\boldsymbol{\epsilon},
\tag{10}
$$

**Algorithm 1** Inferring a point estimate of $p(\mathbf{x}|\mathbf{y}) \approx \delta(\mathbf{x} - \boldsymbol{\eta})$, under a DDPM prior and constraint.

---

**input** pretrained DDPM $\boldsymbol{\epsilon}_\theta$, auxiliary data $\mathbf{y}$, constraint $c$, time schedule $(t_i)_{i=1}^T$, learning rate $\lambda$
 1: Initialize $\mathbf{x} \sim \mathcal{N}(\mathbf{0}; \mathbf{I})$.
 2: **for** $i = T..1$ **do**
 3:     Sample $\boldsymbol{\epsilon} \sim \mathcal{N}(\mathbf{0}; \mathbf{I})$
 4:     $\mathbf{x}_{t_i} = \sqrt{\bar{\alpha}_{t_i}}\mathbf{x} + \sqrt{1 - \bar{\alpha}_{t_i}}\boldsymbol{\epsilon}$
 5:     $\mathbf{x} \leftarrow \mathbf{x} - \lambda\nabla_{\mathbf{x}}[\|\boldsymbol{\epsilon} - \boldsymbol{\epsilon}_\theta(\mathbf{x}_{t_i}, t_i)\|_2^2 - \log c(\mathbf{x}, \mathbf{y})]$
 6: **end for**
**output** $\boldsymbol{\eta} = \mathbf{x}$

---

where the stage $t$ noise reconstruction $\boldsymbol{\epsilon}_\theta(\mathbf{x}_t, t)$ is a linear transformation of the model's expectation $\boldsymbol{\mu}_\theta(\mathbf{x}_t, t)$:

$$\boldsymbol{\mu}_\theta(\mathbf{x}_t, t) = \frac{1}{\sqrt{\alpha_t}}\left(\mathbf{x}_t - \frac{\beta_t}{\sqrt{1 - \bar{\alpha}_t}}\boldsymbol{\epsilon}_\theta(\mathbf{x}_t, t)\right). \tag{11}$$

The weighting $w_t(\beta)$ is generally a function of the noise schedule, but in most pretrained diffusion models it is set to 1. Thus, the free energy in (2) reduces to

$$F = \sum_t \mathbb{E}_{\boldsymbol{\epsilon}\sim\mathcal{N}(\mathbf{0},\mathbf{I})}[\|\boldsymbol{\epsilon} - \boldsymbol{\epsilon}_\theta(\mathbf{x}_t, t)\|_2^2] - \mathbb{E}_{q(\mathbf{x})}[\log c(\mathbf{x}, \mathbf{y})]$$

$$= \sum_t \mathbb{E}_{\boldsymbol{\epsilon}\sim\mathcal{N}(\mathbf{0},\mathbf{I})}[\|\boldsymbol{\epsilon} - \boldsymbol{\epsilon}_\theta(\mathbf{x}_t, t)\|_2^2] - \log c(\boldsymbol{\eta}, \mathbf{y}), \quad \mathbf{x}_t = \sqrt{\bar{\alpha}_t}\boldsymbol{\eta} + \sqrt{1 - \bar{\alpha}_t}\boldsymbol{\epsilon}. \tag{12}$$

The first term is the cost usually used to learn the parameters $\theta$ of the diffusion model. To perform inference under an already trained model $\boldsymbol{\epsilon}_\theta$, *we instead minimize $F$ with respect to $\boldsymbol{\eta}$ through sampling $\boldsymbol{\epsilon}$ in the summands over $t$.*

A similar derivation applies if a Gaussian approximation to the posterior $q(\mathbf{x})$ is used (see §A). Such an approximation allows to model not only a mode of the posterior, but the uncertainty in its vicinity.

We summarize the algorithm for a point estimate $q(\mathbf{x})$ as Algorithm 1. Variations on this algorithm are possible. Depending on how close to a good mode we can initialize $\boldsymbol{\eta}$, this optimization may involve summing only over $t \leq t_{\max} < T$; different time step schedules can be considered depending on the desired diversity in the estimated $\mathbf{x}$. Note that optimization is stochastic and each time it is run it can produce different point estimates of $\mathbf{x}$ which are are both likely under the diffusion prior and satisfy the constraint as much as possible.

We observed that optimizing simultaneously for all $t$ makes it difficult to guide the sample towards a mode in image generation applications; therefore, we anneal $t$ from high to low values. Intuitively, the first few iterations of gradient descent should coarsely explore the search space, while later iterations gradually reduce the temperature to steadily reach a nearby local maximum of $p(\mathbf{x}|\mathbf{y})$. Examples of annealing schedules designed for the tasks demonstrated in §3, 4, 5 are presented in the Appendix (Fig. B.1).

Another interesting case is when $\mathbf{x}$ is parametrized through a latent variable (this can be seen as a case of a hard, non-differentiable constraint: if $\mathbf{x}$ is a deterministic function of $\mathbf{y}$, $\mathbf{x} = f(\mathbf{y})$, then $c(\mathbf{x}, \mathbf{y})$ is supported on the corresponding manifold). Then the procedure in Algorithm 1 can be performed with gradient descent steps with respect to $\mathbf{y}$ on

$$\|\boldsymbol{\epsilon} - \boldsymbol{\epsilon}_\theta(\sqrt{\bar{\alpha}_{t_i}}f(\mathbf{y}) + \sqrt{1 - \bar{\alpha}_{t_i}}\boldsymbol{\epsilon}, t_i)\|_2^2 \tag{13}$$

instead of steps 4 and 5. (For some semantics of the latent representation, one may wish to make the prior on $\mathbf{x}$ the pushforward by $f$ of a known prior on the latent $\mathbf{y}$. In this case, (13) must be weighted by the Jacobian of $f$ at $\mathbf{y}$.)

## 3 Experiments: Conditional image generation

### 3.1 Simple illustration on MNIST

We first explore the idea of generating conditional samples from an unconditional diffusion model on MNIST. We train the DDPM model of [7] on MNIST digits and experiment with different sets of

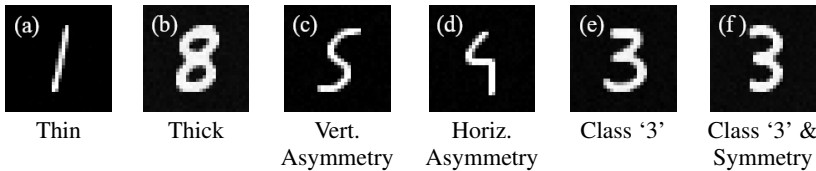

|  |  |  |  |  |  |
|---|---|---|---|---|---|
| Thin | Thick | Vert. Asymmetry | Horiz. Asymmetry | Class '3' | Class '3' & Symmetry |

Figure 1: Inferred MNIST samples under different conditions $c(\mathbf{x}, \mathbf{y})$.

constraints $\log c(\mathbf{x}, \mathbf{y})$ to generate samples with specific attributes. The examples in Fig. 1 showcase such generated samples. For the digit in (a) we set the constraint $\log c$ to be the unnormalized score of 'thin' digits, computed as negative of the average image intensity, whereas in (b) we invert that and generate a 'thick' digit with high mean intensity. Similarly, in (c) and (d) we hand-craft a score that penalizes the vertical and horizontal symmetry respectively, by computing the $L^2$ distance between the two folds (vertical/horizontal) of the digit $\mathbf{x}$, which leads to the generation of skewed, non-symmetric samples.

We also showcase how the auxiliary constraint $c(\mathbf{x}, \mathbf{y})$ can be modeled by a different, independently trained network. The digit in Fig. 1 (e) is generated by constraining the DDPM with a classifier network that is separately trained to distinguish between the digit class $\mathbf{y} = 3$ and all other digits. The auxiliary constraint in this case is the likelihood of the inferred digit, as it is estimated by the classifier. Finally, for (f) we multiply horizontal *symmetry* and digit classifier constraints, prompting the inference procedure to generate a perfectly centered and symmetric digit. Details of model training and inference can be found in the Appendix (§B.1).

## 3.2 Using off-the-shelf components for conditional generation of faces

We consider the generation of natural images with a pretrained DDPM prior and a learned constraint. We utilize the pretrained DDPM network on FFHQ-256 [19] from [3] and a pretrained ResNet-18 face attribute classifier on CelebA [25]. The attribute classifier computes the likelihood of presence of various facial features $y$ in a given image $\mathbf{x}$, as they are defined by the CelebA dataset. Examples of such features are *no beard*, *smiling*, *blond hair* and *male*. To generate a conditional sample from the unconditional DDPM network we select a subset of these and enforce their presence or absence using the classifier predicted likelihoods as our constraint $c$. If $\mathbf{y}$ is a set of attributes we wish to be present, the constraint $\log c(\mathbf{x}, \mathbf{y})$ can be expressed as

$$\log c(\mathbf{x}, \mathbf{y}) = \sum_{y \in \mathbf{y}} \log p(y \mid \mathbf{x}) \tag{14}$$

We only strictly enforce a small subset of facial attributes and therefore $\mathbf{x}$ is allowed to converge towards different modes that correspond to samples that exhibit, in varying levels, the desired features.

In Fig. 2 we demonstrate our ability to infer conditional samples $\mathbf{x}$ with desired attributes $\mathbf{y}$, using only the unconditional diffusion model and the classifier $p(\mathbf{y} \mid \mathbf{x})$. In the first row, we show the results of the optimization procedure of Algorithm 1 for various attributes. The classifier objective $c(\mathbf{x}, \mathbf{y})$ manipulates the image with the goal of making the classifier network produce the desired attribute predictions, whereas the diffusion objective attempts to pull the sample $x$ towards the learned distribution $p(\mathbf{x})$. If we ignored the denoising loss, the result would be some adversarial noise that fools the classifier network. The DDPM prior, however, is strong enough to guide the process towards realistic-looking images that simultaneously satisfy the classifier constraint set.

We notice that the generated samples $\mathbf{x}$, although having converged towards a correct mode of $p(\mathbf{x})$, still exhibit a noticeable amount of noise related to the optimization of classifier objective. To address that, inspired by [31], we simply denoise the image using the DDPM model alone, starting from the low noise level $t = 200$ so as to retain the overall structure. The results of this denoising are shown in the second row of Fig. 2.

In Fig. 3 we showcase the intermediate steps of the optimization process for inference with the conditions *blond hair*+*smiling*+*not male*, thus solving a problem like that studied in [8] using only *independently trained* attribute classifiers and an unconditional generative model of faces. The sample $x$ is initialized with Gaussian noise $\mathcal{N}(\mathbf{0}, \mathbf{I})$, and as we perform gradient steps with decreasing values of $t$, we observe facial features being added in a coarse-to-fine manner.

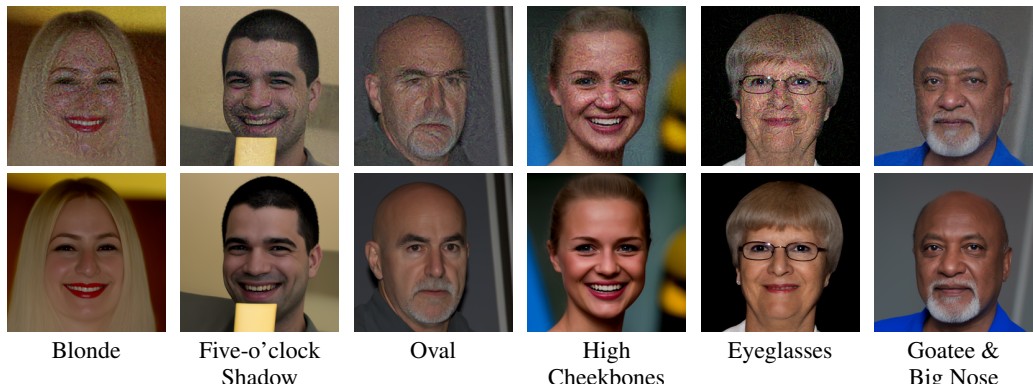

| Blonde | Five-o'clock Shadow | Oval | High Cheekbones | Eyeglasses | Goatee & Big Nose |

Figure 2: First row: Conditional FFHQ samples $\mathbf{x}$ for constraints $c(\mathbf{x}, \mathbf{y})$ with various attribute sets $\mathbf{y}$. Second row: denoising as in [31] to remove artifacts that appear when optimizing with a classifier network enforcing the constraint.

$t = 1000$  $t = 962$  $t = 896$  $t = 807$  $t = 701$  $t = 585$  $t = 465$  $t = 349$  $t = 242$  Denoise

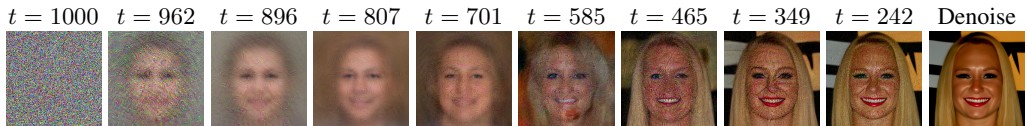

Figure 3: FFHQ conditional generation for $\mathbf{y} = \{Blonde, Smiling, Female\}$. The last step performs denoising as in [31] to remove artifacts that appear when training on a classifier as a constraint.

In the Appendix (§B.2) we provide additional samples and further discuss the sample quality in comparison to unconditional generation. We also present results on inference with conflicting attributes as well as common failure cases.

## 4 Experiments: Semantic image segmentation

We test the applicability of diffusion priors in discrete tasks, such as inferring semantic segmentations from images. For this purpose, we use the EnviroAtlas dataset [32] which is composed of 5-class, 1m-resolution land cover labels from four geographically diverse cities across the US; Pittsburgh, PA, Durham, NC, Austin, TX and Phoenix, AZ. We only have access to the high resolution labels from Pittsburgh, and the task is to infer the land cover labels in the other three cities, given only probabilistic weak labels $\ell_{\text{weak}}$ derived from coarse auxiliary data [34]. We use Algorithm 1 to perform an inference procedure that does not directly take imagery as input, but uses constraints derived from unsupervised color clustering. We use only cluster indices in inference, making the algorithm dependent on image structure, but not color. Local cluster indices as a representation have a promise of extreme domain transferability, but they require a form of a combinatorial search which matches local cluster indices to semantic labels so that the created shapes resemble previously observed land cover, as captured by a denoising diffusion model of semantic segmentations.

**DDPM on semantic pixel labels.** We train a DDPM model on the $\frac{1}{4}$-resolution one-hot representations of the land cover labels, using the U-Net diffusion model architecture from [7]. To convert the one-hot diffusion samples to probabilities we follow [15] and assume that for any pixel $i$ in the inferred sample $\mathbf{x}$, the distribution over the label $\ell$ is, $p(\ell_i) \propto \int_{0.5}^{1.5} \mathcal{N}(x_i^{\ell} \mid \eta_i, \sigma)$, where $\sigma$ is user-defined a parameter. We chose this approach for its simplicity and ease to apply in our inference setting of Algorithm 1. Alternatively, we could use diffusion models for categorical data [14] with the appropriate modifications to our inference procedure. Samples drawn from the learned distribution are presented in Fig 4.

**Inferring semantic segmentations.** In order to infer the segmentation of a single image, under the diffusion prior, we directly apply Algorithm 1 with a hand-crafted constraint $c$ which provides structural and label guidance. To construct $c$, we first compute a local color clustering $\mathbf{z}$ of input

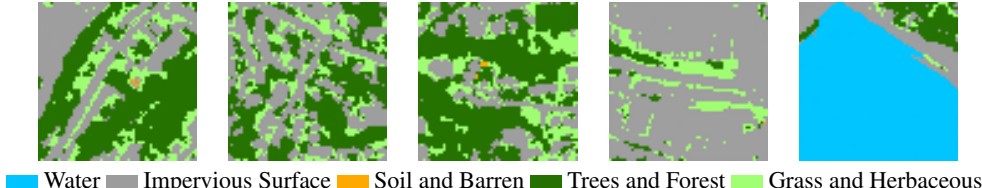

Water ■ Impervious Surface ■ Soil and Barren ■ Trees and Forest ■ Grass and Herbaceous

Figure 4: Unconditional samples from the DDPM trained on land cover segmentations (cf. Fig. 5).

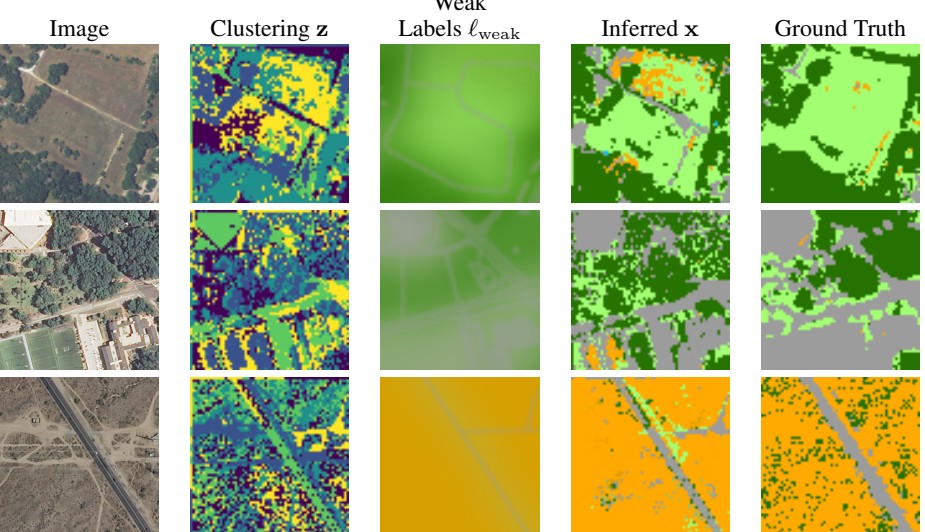

Figure 5: Segmentation inference results. The inferred segmentation $\mathbf{x}$ is initialized with the weak labels to reduce the number of steps needed. The samples are chosen from (top to bottom) Durham, NC, Austin, TX and Phoenix, AZ. Although AZ has a vastly different joint distribution of colors and labels, the inferred segmentation still captures the overall structure. Note that the inference algorithm does not use the pixel intensities in the input image, only an unsupervised color clustering.

the image (§B.3 in the Appendix). In addition, we utilize the available weak labels $\ell_{\text{weak}}$ [34] and force the predicted segments' distribution to match the weak label distribution when averaged in non-overlapping blocks. We combine the two objectives in a single constraint $c(\mathbf{x}, \mathbf{z}, \ell_{\text{weak}})$ by (i) computing the mutual information between the color clustering $\mathbf{z}$ and the predicted labels $\mathbf{x}$, transformed into a valid probability distribution from the inferred one-hot vectors, in overlapping image patches and (ii) computing the negative KL divergence between the average predicted distribution and the distribution given by the weak labels in non-overlapping blocks

$$\log c(\mathbf{x}, \mathbf{z}, \ell_{\text{weak}}) = \text{MI}(\mathbf{x}, \mathbf{z}) - \text{KL}(\mathbf{x} \,\|\, \ell_{\text{weak}}). \tag{15}$$

Empirically, we find that we can reduce the number of optimization steps needed to perform inference by initializing the sample $\mathbf{x}$ with the weak labels $\ell_{\text{weak}}$ instead of random noise, allowing us to start from a smaller $t_i$. Examples of images and their inferred segmentations are shown in Fig. 5.

**Domain transfer with inferred samples.** The above inference procedure is agnostic to colors by design, and we expect it to have a greater ability to perform in new areas than the approach in [34], which still finetunes networks that take raw images as input. We also investigate domain transfer approaches where patches segmented using the the diffusion prior are used to train neural networks for fast inference. We pretrain a standard U-Net inference network $p(\mathbf{x} \mid I)$ solely on 20k batches of 16 randomly sampled $64 \times 64$ image patches in PA. We randomly sample 640 images in each of the other geographies and generate semantic segmentations using our inference procedure, then finetune the inference network on these segmentations. This network is then evaluated on the entire target geography.

Table 1: Accuracies and class mean intersection-over-union scores on the EnviroAtlas dataset in various geographic domains. The model in the second-to-last row was pretrained in a supervised way on labels in the Pittsburgh, PA, region.

| Algorithm | Durham, NC | | Austin, TX | | Phoenix, AZ | |
|---|---|---|---|---|---|---|
| | Acc % | IoU % | Acc % | IoU % | Acc % | IoU % |
| PA supervised | 74.2 | 35.9 | 71.9 | 36.8 | 6.7 | 13.4 |
| PA supervised + weak | 78.9 | 47.9 | 77.2 | 50.5 | 62.8 | 24.2 |
| Implicit posterior [34] | 79.0 | 48.4 | 76.6 | 49.5 | 76.2 | 46.0 |
| Ours (from scratch) | 76.0 | 39.9 | 74.8 | 39.4 | 69.5 | 31.6 |
| Ours (fine-tuned) | 79.8 | 46.4 | 79.5 | 45.4 | 69.6 | 32.4 |
| Full US supervised [33] | 77.0 | 49.6 | 76.5 | 51.8 | 24.7 | 23.6 |

The results in Table 1 demonstrate that this approach to domain transfer is comparable with the state-of-the-art work of [34] for weakly-supervised training. The naïve approach of training a U-Net only on the available high-resolution PA data (PA supervised) fails to generalize to the geographically different location of Phoenix, AZ. Similarly, the model of [33], which is a US-wide high-resolution land cover model trained on imagery and labels, and multi-resolution auxiliary data over the entire contiguous US also suffers. When the weak labels are provided as input (PA supervised + weak) the results can improve significantly.

## 5 Experiments: Continuous relaxation of combinatorial problems

So far, we have considered inference under a DDPM prior and a differentiable constraint $c(\mathbf{x}, \mathbf{y})$. We consider the case of a 'hard' constraint, where $\mathbf{y}$ is a latent vector deterministically encoded in an image $\mathbf{x}$ ($\mathbf{x} = f(\mathbf{y})$) and we have a DDPM prior over images $p_{\text{DDPM}}(\mathbf{x})$. We will use the variation of Algorithm 1 described at the end of §2.2 to obtain a point estimate of the distribution over $y$, $p(\mathbf{y}) \propto p_{\text{DDPM}}(f(\mathbf{y}))$.

We illustrate this in the setting of a well-known combinatorial problem, the traveling salesman problem (TSP). Recall that a Euclidean traveling salesman problem on the plane is described by $N$ points $v_1, \ldots, v_N \in \mathbb{R}^2$, which form the vertex set of a complete weighted graph $G$, where the weight of the edge from $v_i$ to $v_j$ is the Euclidean distance $\|v_i - v_j\|$. A *tour* of $G$ is a connected subgraph in which every vertex has degree 2. The TSP is the optimization problem of finding the tour with minimal total weight of the edges, or, equivalently, a permutation $\sigma$ of $\{1, 2, \ldots, N\}$ that minimizes

$$\|v_{\sigma(1)} - v_{\sigma(2)}\| + \|v_{\sigma(2)} - v_{\sigma(3)}\| + \cdots + \|v_{\sigma(N-1)} - v_{\sigma(N)}\| + \|v_{\sigma(N)} - v_{\sigma(1)}\|.$$

Although the general form of the TSP is NP-hard, a polynomial-time approximation scheme is known to exist in the Euclidean case [2, 28] and can yield proofs of tour optimality for small problems.

Humans have been shown to have a natural propensity for solving the Euclidean TSP (see [26] for a survey). Humans construct a tour by processing an image representation of the points $v_1, \ldots, v_N$ through their visual system. However, the optimization algorithms in common use for solving the TSP do not use a vision inductive bias, instead falling into two broad categories:

- Discrete combinatorial optimization algorithms and efficient integer programming solvers, studied for decades in the optimization literature [24, 12, 10];
- More recently, there has been work on neural nets, trained by reinforcement learning or imitation learning, that build tours sequentially or learn heuristics for their (discrete) iterative refinement. Successful recent approaches [6, 23, 16, 17, 4] have used Transformer [44] and graph neural network [22] architectures.

The algorithm we propose using DDPMs is a hybrid of these categories: it reasons over a continuous relaxation of the problem, but exploits the learning of generalizable structure in example solutions by a neural model. In addition, ours is the first TSP algorithm to mimic the convolutional inductive bias of the visual system.

| | Optimize latent adjacency matrix w.r.t. denoising model | | | | | Recover tour | | |
|---|---|---|---|---|---|---|---|---|

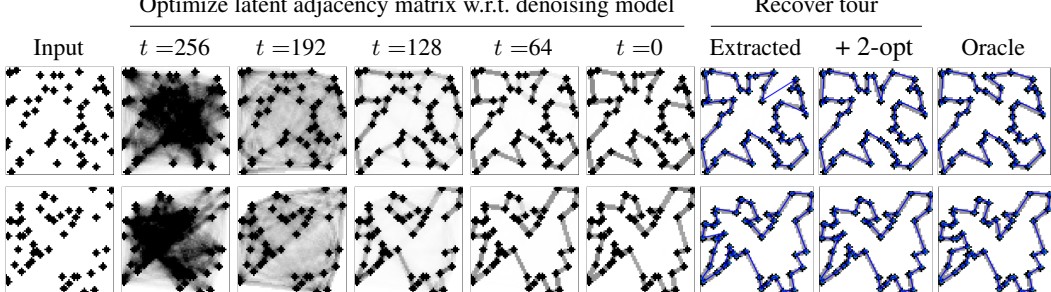

| Input | $t=256$ | $t=192$ | $t=128$ | $t=64$ | $t=0$ | Extracted | + 2-opt | Oracle |

Figure 7: The procedure for solving the Euclidean TSP with a DDPM: Gradient descent is performed on a latent adjacency matrix $A$ to minimize a stochastic denoising loss on an *image representation* $f(A)$ with steadily decreasing amounts of noise (here, 256 steps). In the process, pieces of the tour are 'burned in' and later recombined in creative ways. Finally, a tour is extracted from the inferred adjacency matrix and refined by uncrossing moves. For both problems shown, the length of the inferred tour is within 1% of the optimum.

**Encoding function.** Fix a set of points $v_1, \dots, v_N \in [0,1] \times [0,1]$. We encode an symmetric $N \times N$ matrix with 0 diagonal $A$ as a $64 \times 64$ greyscale image $f(A)$ by superimposing: (i) raster images of line segments from $v_i$ to $v_j$ with intensity value $A_{ij}$ for every pair $(i,j)$, and (ii) raster images of small black dots placed at $v_i$ for each $i$. For example, if $A$ is the adjacency matrix of a tour, then $f(A)$ is a visualization of this tour as a $64 \times 64$ image.

**Diffusion model training.** We use a dataset of Euclidean TSPs, with ground truth tours obtained by a state-of-the-art TSP solver [10], from [23] (we consider two variants of the dataset, each with $\sim$1.5m training graphs: with 50 vertices in each graph and with a varying number from 20 to 50 vertices in each graph). Each training tour is represented via its adjacency matrix $A$ and encoded as an image $f(A)$. We then train a DDPM with the U-Net architecture from [7] on all of such encoded image. Model and training details can be found in the Appendix (§B.4). Some unconditional samples from the trained DDPM are shown in Fig. 6; most samples indeed resemble image representations of tours.

**Solving new TSPs.** Suppose we are given a new set of points $v_1, \dots, v_N$. Solving the TSP requires finding the adjacency matrix $A$ of a tour of minimal length. As a differentiable relaxation, we set $A = S + S^\top$, where $S$ is a stochastic $N \times N$ matrix with zero diagonal (parametrized via softmax of a matrix of parameters over rows). We run the inference procedure using the trained DDPM $p_{\text{DDPM}}(f(A))$ as a prior to estimate $A$. The hyperparameters and noise schedule are described in §B.4. Examples of the optimization are shown in Fig. 7.

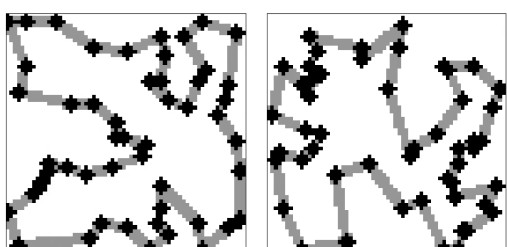

Figure 6: Two unconditional samples from the diffusion model trained on images of solved TSPs.

Although the inferred $A$ is usually sharp (i.e., all entries close to 0 or 1), rounding $A$ to 0 or 1 does not always give the adjacency matrix of a tour (see, for example, the top row of Fig. 7; other common incorrect outputs include pairs of disjoint tours). To extract a tour from the inferred $A$, we greedily insert edges to form an initial proposal, then refine it using a standard and lightweight combinatorial procedure, the 2-opt heuristic [24] (amounting to iteratively uncrossing pairs of edges that intersect). The entire procedure is shown in Fig. 7, and full details can be found in the Appendix (§B.4).

**Results.** We evaluate the trained models on test sets of 1280 graphs each with $N = 50$ and $N = 100$ vertices. We report the average length of the inferred tour and the gap (discrepancy from the length of the ground truth tour) in Table 2 (left), from which we make several observations.

Table 2: *Left:* Mean tour length and optimality gap on Euclidean TSP test sets. The baseline results from [23, 16, 4] are taken from the respective papers. The two DDPMs were trained on 1.5m images of solved TSP instances (with different numbers of vertices) and used to infer latent adjacency matrices in the test set. *Right:* Performance of the DDPM trained on images of 50-vertex TSP instances with different numbers of inference steps (see the Appendix (§B.4) for time schedule details). We also show the mean number of 2-opt (uncrossing) steps per instance, suggesting that the DDPM prior assigns high likelihood to adjacency matrices that are in less need of refinement.

| Algorithm | $N = 50$ | | $N = 100$ | |
|---|---|---|---|---|
| | Obj | Gap % | Obj | Gap % |
| Oracle (Concorde [10]) | 5.69 | 0.00 | 7.759 | 0.00 |
| 2-opt [24] | 5.86 | 2.95 | 8.03 | 3.54 |
| Transformer [23] | 5.80 | 1.76 | 8.12 | 4.53 |
| GNN [16] | 5.87 | 3.10 | 8.41 | 8.38 |
| Transformer [4] | 5.71 | 0.31 | 7.88 | 1.42 |
| Diffusion 20–50 | 5.76 | 1.23 | 7.92 | 2.11 |
| Diffusion 50 | 5.76 | 1.28 | 7.93 | 2.19 |

| Diff. steps | $N = 50$ | | | $N = 100$ | | |
|---|---|---|---|---|---|---|
| | Obj | Gap % | Steps | Obj | Gap % | Steps |
| 256 | 5.763 | 1.28 | 11.6 | 7.930 | 2.19 | 50.6 |
| 64 | 5.780 | 2.60 | 14.3 | 7.942 | 2.35 | 45.7 |
| 16 | 5.858 | 2.98 | 25.9 | 8.052 | 3.78 | 58.6 |
| 4 | 5.851 | 2.86 | 23.9 | 8.031 | 3.50 | 52.8 |
| 2-opt | 5.856 | 2.95 | 24.4 | 8.034 | 3.54 | 53.0 |

- The right side of Table 2 shows the number of 2-opt (edge uncrossing) steps performed in the refinement step of the algorithm when the inference algorithm is run for varying numbers of steps. Running the inference with more steps results in extracted tours that are closer to local minima with respect to the 2-opt neighbourhood, indicating that the DDPM encodes meaningful information about the shape of tours.

- The DDPM inference is competitive with recent baseline algorithms that do not use beam search in generation of the tour (those shown in the table). These baseline algorithms improve when beam search decoding with very large beam size is used, but encounter diminishing returns as the computation cost grows. Our performance on the 100-vertex problems is similar to [23] with the largest beam size they report (5000), which has 2.18% gap, while having similar computation time.

- The model trained on problems with 50 nodes performs almost identically to the model trained on problems with 50 or fewer nodes, and both models generalize better than baseline methods from 50-node problems to the out-of-distribution 100-node problems.

We emphasize a unique feature of our algorithm: all 'reasoning' in our inference procedure happens via the image space. This property also leads to sublinear computation cost scaling with increasing size of the graph – as long as it can reasonably be represented in a $64 \times 64$ image – since most of the computation cost of inference is borne by running the denoiser on images of a fixed size. In the Appendix (§B.4) we explore the generalization of the model trained on 20- to 50-node TSP instances to problems with 200 nodes and discuss potential extensions.

# 6 Conclusion

We have shown how inference in denoising diffusion models can be performed under constraints in a variety of settings. Imposing constraints that arise from pretrained classifiers enables conditional generation, while common-sense conditions, such as mutual information with a clustering or divergence from weak labels, can lead to models that are less sensitive to domain shift in the distribution of conditioning data.

A notable limitation of DDPMs, which is inherited by our algorithms, is the high cost of inference, requiring a large number of passes through the denoising network to generate a sample. We expect that with further research on DDPMs for which inference procedures converge in fewer steps [37, 45], plug-and-play use of DDPMs will become more appealing in various applications.

Finally, our results on the traveling salesman problem illustrate the ability of DDPMs to reason over uncertain hypotheses in a manner that can mimic human 'puzzle-solving' behavior. These results open the door to future research on using DDPMs to efficiently generate candidates in combinatorial search problems.

## Acknowledgments

The authors thank the anonymous NeurIPS 2022 reviewers for their comments.

All authors are funded by their primary institutions. Partial support was provided by the NASA Biodiversity program (Award 80NSSC21K1027), NSF grants IIS-2123920 and IIS-2212046, and the Partner University Fund 4D Vision award.

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
