# OpenReview forum: "Diffusion Models as Plug-and-Play Priors"
_NeurIPS.cc/2022/Conference — NeurIPS 2022 Accept_

### Official Review · Reviewer_akx3 · 2022-07-10

**Rating:** 5
**Confidence:** 3
**Soundness:** 3 good
**Presentation:** 2 fair
**Contribution:** 2 fair

**Summary:**

This paper uses the diffusion model as prior modeling, and it mainly focuses on downstream tasks.

**Questions:**

-

**Limitations:**

-

**Strengths And Weaknesses:**

$\textbf{Strengths}$
- This paper formalizes the estimation of the posterior distribution using a pre-trained diffusion model. This is not a new idea, but its strength lies in the formalization and vast applications.
- A majority of conditional diffusion models re-train their score network, but having a fixed diffusing SDE, there is no need to re-train the estimated perturbed score function, and an endeavor is needed to solve downstream tasks without re-training the pre-trained diffusion model. This paper gives the most general framework in this aspect.
- Aside from the framework, the novelty of this paper could be evaluated by its application cases. It solves semantic image segmentation and traveling salesman problem using DDPM, which is the first to the best of my knowledge. Solving these problems with DDPM is a significant novelty.

$\textbf{Weaknesses}$
- The framework is not a new idea, and a number of papers already applied this framework without mentioning or formalizing explicitly. Therefore, suggesting a framework itself is relatively a weak contribution in the community.
- We think that the authors suggested a framework on an abstract level because they tried to put multiple downstream tasks in a single paper. However, generally, this is not a desirable approach to write a paper. We recommend authors to focus down on each task in separate manuscripts.

---

> ### Author Response · Authors · 2022-07-29
> **Author response**
>
> Thank you for the comments. We respond to them in order below.
>
> > The framework is not a new idea, and a number of papers already applied this framework without mentioning or formalizing explicitly. Therefore, suggesting a framework itself is relatively a weak contribution in the community.
>
> Please see the response to all reviewers, where we clarify the difference between our approach (optimization) and conditional diffusion methods (sampling). If you believe we have missed some coinciding recent work, we kindly ask you to point us to it.
>
> > We think that the authors suggested a framework on an abstract level because they tried to put multiple downstream tasks in a single paper. However, generally, this is not a desirable approach to write a paper. We recommend authors to focus down on each task in separate manuscripts.
>
> The idea we are trying to get across is that DDPMs can be utilized as priors and be flexibly applied to very diverse settings, including those where generative models of images have not been used in the past. Our main focus was to demonstrate how thinking of DDPMs as priors help tackle problems that were previously solved by conditional DDPMs (image generation, segmentation) and as a novel solution to yet-unexplored problems (domain transfer, combinatorial optimization).

---

### Official Review · Reviewer_6Fox · 2022-07-11

**Rating:** 5
**Confidence:** 3
**Soundness:** 3 good
**Presentation:** 3 good
**Contribution:** 3 good

**Summary:**

In this work, the authors consider the problem of inferring high-dimensional data x in a model that consists of a prior and an auxiliary constraint, which has a differentiable form and can come from diverse sources. The authors demonstrate the efficiency of the proposed framework in different tasks.

**Questions:**

Please refer to the weaknesses part.

**Limitations:**

The author addressed some limitations in the conclusion part.

**Strengths And Weaknesses:**

Strengths:
- Interesting idea. The proposed model provides different conditional information, which seeks samples x that satisfy both the classifier and the prior. It does not require additional training or finetuning model components or inference networks.
- Novel methodology. Though recently using diffusion priors for high-fidelity synthesis has been studied in a wide range of literature, the proposed algorithm in this work is original and novel.
- Good presentation. The paper is well-written.


Weaknesses:
- Insufficient experiments. The authors conduct experiments on conditional generation, image segmentation, and combinatorial search problems. However, quantitative analysis is missing. How does the proposed model outperform previous arts? Furthermore, the author spends two pages describing the traveling salesman's problems, while the reviewer thinks focusing more on the benchmark image synthesis task could be better.
- More Question: The author presents their classifier-guided conditional image synthesis results on the FFHQ dataset, but what is the difference between some popular Guided-Diffusion methods?

---

> ### Author Response · Authors · 2022-07-29
> **Author response**
>
> Thank you for your comments. We respond to them in order below.
>
> > Insufficient experiments. The authors conduct experiments on conditional generation, image segmentation, and combinatorial search problems. However, quantitative analysis is missing. How does the proposed model outperform previous arts? Furthermore, the author spends two pages describing the traveling salesman's problems, while the reviewer thinks focusing more on the benchmark image synthesis task could be better.
>
> We focused our quantitative analyses on the image segmentation and combinatorial optimization problems, which are the domains to which unconditional DDPMs have never been applied before. Although a quantitative analysis is not provided for the conditional image generation task, it is presented to showcase qualitatively how the proposed algorithm can be used to solve problems already examined by past diffusion papers, typically using **conditional** DDPMs.
>
> In our conditional image generation experiments, we used a pretrained unconditional DDPM trained on the FFHQ dataset and combined it with a face attribute classifier trained independently on the CelebA dataset: paired images were never seen. Naturally, models trained jointly on images and labels would perform best, but the message of our paper is that our algorithm can produce realistic conditional samples without being trained for conditional generation.
>
> > More Question: The author presents their classifier-guided conditional image synthesis results on the FFHQ dataset, but what is the difference between some popular Guided-Diffusion methods?
>
> First, please see the response to all reviewers.
>
> More specifically, in recent works, such as [7] and [5], the authors presented algorithms to 'guide' the diffusion process towards a desired subset of images (e.g., people with glasses or blond-haired people) by modifying the sampling algorithm steps $p_{theta}(x_{t-1} \mid x_t)$. Our approach does away completely with the DDPM sampling procedure and poses conditional sampling as an optimization process; we directly optimize the 'denoised' image $x_0$ such that it satisfies the DDPM prior **and** an additional auxiliary constraint. This allows us to fully decouple the training process of the diffusion model from the constraints (which may be pretrained or fixed differentiable functions). For example, [7] required training the attribute classifier $p(y \mid x_t)$ with the *noisy* images $x_t$, whereas we use an independently trained classifier of *non-noisy* images, trained on a different dataset than the DDPM, to generate samples with specific attributes.

---

### Official Review · Reviewer_3SJz · 2022-07-12

**Rating:** 6
**Confidence:** 2
**Soundness:** 3 good
**Presentation:** 3 good
**Contribution:** 3 good

**Summary:**

This paper proposed to use pre-trained denoising diffusion probabilistic models (DDPM) as priors for other inference tasks. The way the authors use DDPM is to revert a DDPM by inputting $x$ and output a random noise $\epsilon$. By comparing the difference between inverse mapping and noise (usually Gaussian), the proposed framework can ensure $x$ lie in the range of DDPM. Furthermore, the authors construct an optimization problem where the loss function is the summation of log likelihood (auxiliary constraint) and l2 loss. By iteratively searching for the solution that both stays in the range of DDPM and satisfies the constraint, the proposed framework makes DDPM a plug-and-play prior for arbitrary inference tasks.

**Questions:**

1. Please show more detailed deriviation from Eq.7-11, especially the re-parameterization trick.
2. line 109, "To perform nference under an already trained model ϵθ, we instead minimize F wrt to η (and ψ if desired) through sampling ϵ in the summands over t.". Don't you directly minimize $x_t$?
3. In the segmentation task, how were the DDPM networks finetuned? Does scratch mean pre-trained?

**Limitations:**

One limitation of the experiment is the lack of image restoration tasks, which I believe can better demonstrate the prior information provided by DDPM.

**Strengths And Weaknesses:**

**Strength:**
- The author proposes a novel Plug-and-play priors by leveraging DDPM. Different from the existing priors based on GAN, the proposed framework inversely map the image $x$ into the latent space. In this way, the iterate is no longer the latent variable but the actual image.
- The paper demonstrates its generality on a wide range of tasks, including segmentation, face generation, and traveling salesman problems. It is my first time to see someone solves a combinatorial problem by using DDPM.

**Weakness:**
- The paper is not easy to read. There is not enough detailed derivations between eq.7-11
- The novelty of the work is not clearly stated.
- The naming of the framework is confusing. Plug-and-play priors (PnP) is a well-known framework in the literature of image restoration and imaging inverse problems. The title confuses me in the first place.

---

> ### Author Response · Authors · 2022-07-29
> **Author response**
>
> Thank you for the comments. We respond to them in order below.
>
> > One limitation of the experiment is the lack of image restoration tasks, which I believe can better demonstrate the prior information provided by DDPM.
>
> We thank the reviewer for the suggestion and agree that image restoration and inverse problems would indeed be interesting to try our methodology on. We believe that the novelty lies in our ability to solve a diverse set of problems, which we demonstrated with our experiments on conditional image generation, image segmentation and combinatorial optimization.
>
> > The novelty of the work is not clearly stated.
>
> Please see the response to all reviewers, where we clarify the novelty of our work: formulating conditional sampling from an unconditional DDPM as an optimization problem and showcasing our unified approach under diverse and novel settings. The main message of the paper is that by optimizing the DDPM objective w.r.t. the sample $x$ (or a latent representation of it), we can utilize the prior knowledge encoded in the DDPM for downstream tasks.
>
> > The naming of the framework is confusing. Plug-and-play priors (PnP) is a well-known framework in the literature of image restoration and imaging inverse problems. The title confuses me in the first place.
>
> The naming is inspired by [27], where a similar idea (unrelated to imaging inverse problems) was shown for Generative Adversarial Networks; we regret the clash of terminology, but we did not introduce it. (Please also see the last section of the response to Reviewer FMow.)
>
> > Please show more detailed deriviation from Eq.7-11, especially the re-parameterization trick.
>
> The derivation is analogous to the one done in [13] for $\beta_0 = \psi$. We would happily add it in detail when we have more space.
>
> > Line 109, "To perform inference under an already trained model $\epsilon_{\theta}$, we instead minimize $F$ wrt to $\eta$ (and $\psi$ if desired) through sampling $\epsilon$ in the summands over t.". Don't you directly minimize $x_t$?
>
> We are directly minimizing an objective w.r.t. the parameters of the posterior over $x$ ($\eta$ and $\psi$). This objective involves an expectation over $\epsilon$, and we are estimating its gradient through sampling of $\epsilon$. Additionally, it is also a summation over all $t$, but, similarly to how DDPMs are trained, we only optimize for a single timestep at a time. However, unlike in DDPM training, the order of $t$ is not randomized but rather predefined by a schedule. We apologize if the notation in Algorithm 1 made it misleading.
>
> > In the segmentation task, how were the DDPM networks finetuned? Does scratch mean pre-trained?
>
> The DDPM network was only trained on the high-res labels from one area (PA) and **not finetuned in the new domains**. Our assumption is that, among the varying geographies, the spatial structure of labels does not change as significantly as the distribution of images. Using the DDPM that has learned the distribution of labels in PA, we are able to perform inference (assisted by the weak supervision) in any other area and finetune the color-based networks that otherwise completely fail at generalizing from one geographic region to another.

---

> > ### Comment · Reviewer_3SJz · 2022-08-08
> > **Thank you**
> >
> > Thank you for the response. I have no further questions and believe that *weak acceptance* is a proper recommendation for the work.

---

### Official Review · Reviewer_FMow · 2022-07-12

**Rating:** 5
**Confidence:** 3
**Soundness:** 3 good
**Presentation:** 2 fair
**Contribution:** 3 good

**Summary:**

The authors propose a new conditional sampling approach by using denoising diffusion models (DDPM) as priors, in which they explicitly model the auxiliary constraint (e.g., pretrained classifiers, etc.) in the reverse diffusion process. The resulting model is demonstrated on conditional image generation, semantic image segmentation, and traveling salesman problem (TSP). Experiments show that their new conditional sampling model can be implemented in practice.

**Questions:**

1), The instruction of the proposed method in Sec. 2.2 is somehow ambiguous and hard to follow in some parts. Specifically, in line 97-98, what’s the value or definition of $\eta$ and $\psi$ and how to link $q(x_t)$ in Eq. 8 to the tractable forward process $q(x_t | x_0) = N(x_t;\sqrt{\bar{a_t}}x_0,(1-\bar{a_t})\mathsf{I})$ in [Jonathan. et al, 2020]. It is also unclear the meaning of $ q(x_{t-1} | x_t, \eta, \psi) $ in the KL in Eq. 9. Is this the same as $ q(x_{t-1} | x_t, x_0) $? Additionally, what is the objective function F minimizing?  In brief, a clearer explanation and instruction of the proposed method will be helpful.

2), From very high-level point of view, what make this proposal different from existing score-based conditional model such as [R1, R2]? The existing methods can also take pretrained unconditional generative models as denoising priors to solve cross domain tasks such as image inverse problems, which is similar to the “plug-and-play” term used in this work.

3), In the same line above, what is the main difference between GAN-based priors and the diffusion model priors? Will the predefined time schedule affect the generation performance? A deeper understanding and intuition would be helpful.

4), There lack enough numerical comparisons to baseline methods for conditional image generation. How is the performance compared to other SOTA conditional generative models? Only visual examples of the proposed method are presented in current manuscript.

Ref:

[R1] Song, et al., Solving Inverse Problems in Medical Imaging with Score-Based Generative Models, 2021.

[R2] Kawar, et al., SNIPS: Solving Noisy Inverse Problems Stochastically, 2021.

Minor comments:

Typos in line 33, 237-238, and 266-267 in the main manuscript.


**Limitations:**

The authors mentioned the high computational cost by using DDPMs as priors to generate samples. This is also the bottleneck for many DDPMs.

**Strengths And Weaknesses:**

This paper builds on other works in the recent literature and proposes something useful. It is overall well written. It presents a conditional generation model that can take many pretrained unconditional DDPMs as “black-box” prior. The numerical results demonstrate the effectiveness of this proposal on various imaging and none imaging tasks. However, the writing of the method section could be improved to make it easier to follow. Additionally, the methodological contributions and novelty of this proposal is somewhat unclear due to the lack of literature review and a deeper comparison to existing work. Please find my technical comments as follows.

---

> ### Author Response · Authors · 2022-07-29
> **Author response**
>
> Thank you for the comments. We respond to them in order below.
>
> >  The instruction of the proposed method in Sec. 2.2 is somehow ambiguous and hard to follow in some parts. Specifically, in line 97-98, what’s the value or definition of $\eta$ and $\psi$...
>
> We chose the approximation to the posterior over $x_0$ to be a Gaussian ${\cal N}(\eta,\psi)$ (eq. 7) because it yields a simple expression for the posterior over $x_t$ at an arbitrary time step (8), which closely resembles the expression in the original DDPM formulation. Although we derive our algorithm for a Gaussian posterior, in practice we take  $\psi\to0$ and are only interested in the point estimate $\eta$.
>
> As the use of a Gaussian posterior only seems to have introduced confusion, we can rewrite this section to use a point estimate of the posterior, i.e., the procedure in Algorithm 1 and in our experiments, and relegate discussion of a Gaussian posterior to the Appendix.
>
> > Additionally, what is the objective function F minimizing?
>
> By minimizing $F$ we minimize the divergence between our approximation to the posterior $q(x) = {\cal N}(\eta,\psi)$ and the true posterior (cf. L71).
>
> > From very high-level point of view, what make this proposal different from existing score-based conditional model such as [R1, R2]? The existing methods can also take pretrained unconditional generative models as denoising priors to solve cross domain tasks such as image inverse problems, which is similar to the “plug-and-play” term used in this work.
>
> First, please see the response to all reviewers, which clarifies the difference between our approach and recent work. Specifically, the papers mentioned fall in the category of modifications to the DDPM sampling algorithm to generate conditional samples, under some assumptions on the conditions. We again emphasize that we **do away completely with the sampling algorithm** of the DDPM and instead pose conditional generation as an **optimization problem**.
>
> Second, we did not intend for 'plug-and-play' to clash with the terminology related to cross-domain tasks. In fact, the term has also been used in works like [27] ("Plug & Play Generative Networks...") to refer to in-domain generation under differentiable constraints, which is the sense we intended.
>
> > What is the main difference between GAN-based priors and the diffusion model priors?
>
> A GAN-based prior, as in [27], has to operate on a latent space (e.g., the input space of the generator network). In contrast, we are free to directly operate both on the image space (conditional image generation, image segmentation) or a latent representation (traveling salesman) without added overhead. For example, our approach to the TSP -- optimizing the graph structure directly through a differentiable mapping from the adjacency matrix to the image space -- would not be possible with a GAN: the latent space of the GAN that generates images is different from the latent space of adjacency matrices in which the optimization happens.
>
> > Will the predefined time schedule affect the generation performance?
>
> The time schedule controls the granularity of the information we are searching for at each step. We designed the time schedules empirically (see the Appendix), having in mind a coarse-to-fine process. We think that determining the optimal schedule for each individual task is an interesting direction to explore in the future. (Note that efficient schedules are also a subject of ongoing research in DDPM **training** and unconditional sampling.)
>
> > There lack enough numerical comparisons to baseline methods for conditional image generation.
>
> The purpose of the conditional image generation experiments is to showcase qualitatively how our proposed method can solve a problem that is familiar to the DDPM community. Since our novelty lies in the ability to apply the DDPM as a prior in diverse settings, we focused our quantitative analysis on the image segmentation and combinatorial optimization tasks, which have not been solved before in this manner.
>
> Compared to conditional generative models that have been trained on a dataset of **paired** images with conditions, our approach will naturally perform worse, since we only combine the two sources of information during inference. We emphasize that in practice, this 'plug-and-play' property -- the flexibility to combine images and conditions from completely decoupled sources -- is our main strength. We showcase that in our conditional image generation experiments: the DDPM prior over images was trained on a different dataset than the network classifying the facial attributes (FFHQ and CelebA, respectively). That is, images paired with their attribute labels were never seen in training!
>
> One can investigate further the optimal ways of jointly training the two networks, such that the quality of the results would compete with the state of the art, but we believe that it would not add to the argument of the present paper.

---

> > ### Comment · Reviewer_FMow · 2022-08-08
> > **Post-Rebuttal Comment**
> >
> > The reviewer would like to first thank the authors for their great efforts in both revising the manuscript and providing a conclusive response. The authors put great effort into addressing my concerns. However, the novelty differences between this proposal and the existing conditional score/denoising diffusions are still unclear as its current state. The reviewer also agree with other reviewers that this proposal should at least provides more comparisons on imaging problems where the original diffusion modes were proposed to solve for.
> >
> > As a result, I think this work is on the borderline of this year's NeurIPS and I would like to maintain my borderline acceptance.

---

> > > ### Author Response · Authors · 2022-08-09
> > > **Thank you + clarifications**
> > >
> > > Thanks for your constructive comments.
> > >
> > > Regarding novelty, we should emphasize three points:
> > >
> > > - Unlike *most* conditional diffusion models, our method does not need a model to be trained on new paired samples, but rather it stitches two different separately defined (and usually preexisting) components: a diffusion model and a constraint of interest. The conditional generation is performed in a very different manner, through a search for a likely output  $x$ of the reverse diffusion process which satisfies the constraint. We show that this search is efficient with gradient descent on $x$ due to the hierarchical structure of diffusion models. We further show that some of the problems that traditional conditional diffusion models cannot be used for, our approach can.
> > > - There are specific problems where models do not need to be trained on paired data, including your suggested references [R1,R2]. Although these approaches have similarities with ours, they assume a **linear** inverse problem, while we work with arbitrary differentiable constraints.
> > > - In addition, our approach is more general as we do not need to traverse the entire chain of timesteps in order when denoising, and in fact we can infer $x$ in a variety of regimes, including non-monotonic noise schedules. We hope future work (by us and others) would explore regimes that "heat" and "cool" to avoid local minima (cf. Figure B.1 in the Appendix); we are especially excited about potential to use these ideas to search for solutions to NP-Hard problems like TSP.

---

### Author Response · Authors · 2022-07-29
**Response to all reviewers: Differences with conditional & guided diffusion**

We would like to thank all the reviewers for their comments and suggestions.

We want to point out the main differences between our approach and those mentioned by reviewers as similar works. In summary, past work on conditional generation with DDPMs either needed training as such (on paired images and labels) or ad hoc adaptation at sampling time. We show that it is possible to do inference by minimizing an optimization criterion that combines an independently trained unconditional DDPM with a constraint.

### Differences with conditional diffusion models

Conditional diffusion models have to be trained as such: with paired image and attribute data. Our algorithm assumes a pretrained **unconditional** DDPM and performs inference under an arbitrary differentiable condition, as illustrated in Sections 3.1 and 3.2. This allows us to tackle a variety of different problems with a unified approach, without enforcing any assumptions regarding the structure of the samples or conditions.

### Differences with guided diffusion / modified sampling from DDPMs

We are posing the task of conditional sampling from an unconditional DDPM model as an **optimization** problem (minimizing the free energy $F$), not a correction or adjustment step added to the DDPM **sampling** algorithm.
**This formulation makes it possible to solve problems using DDPMs that cannot be solved using modified sampling**: apart from conditional image generation, which has been the focus of many past works on diffusion models, we also showcase our ability to solve a segmentation and a combinatorial optimization problem -- all under the same framework.
- In our land cover segmentation experiments, we argue that a model trained on pairs of images and segmentations would perform poorly due to domain transfer. Our approach only assumes that the prior over labels, modeled by a DDPM, is domain-invariant, and accounts for the change in correspondence of appearances to labels by tuning image-to-label networks with a small set of segmentations obtained using our algorithm from the DDPM. This is different from what conditional DDPMs can do in two ways:
   - The final output is a domain-specific image-to-label network that can be efficiently used on new data, rather then an output generated by the diffusion model itself.
   - Domain-specific training data was never seen by the DDPM: the DDPM was trained only on data that is assumed to be domain-invariant (just the spatial structure of labels, not the correspondence of colors to labels).
- For the traveling salesman problem, our method optimizes the latent **graph** structure directly by 'fitting' the corresponding graph image to a DDPM prior over images of TSP solutions. In contrast, conventional (and conditional) diffusion algorithms are not able to optimize with respect to a latent when the DDPM is trained on a **representation** of this latent.

---

### Author Response · Authors · 2022-08-08
**Questions after author response?**

Dear reviewers,

The end of the discussion phase is approaching, and we would like to ask if you have any additional questions for us after reading our responses.

Thank you,

The authors.

---

### Meta-Review · Area_Chair_3b7K · 2022-08-24

**Recommendation:** Accept
**Confidence:** Less certain

**Metareview:**

Ratings: 5/5/6/5.
Confidence: 3/3/2/3.
Discussion among reviewers: No.

Summary: This paper proposes a framework for solving conditional image generation tasks through an optimization procedure where a pre-trained diffusion model p(x) is combined with a separate constraint c(x,y). The method is applied to conditional image generation, image segmentation and the traveling salesman problem.

Although the reviewers maintained ratings of accept/borderline accept, the authors wrote detailed responses to the reviewers' concerns, revised the manuscript, and actively engaged in discussion. There is rising interest in diffusion models, largely due to their state-of-the-art image quality results, and this paper will probably be of interest to the NeurIPS community. My recommendation is to accept.

**Award:**

No

---

### Decision · Program_Chairs · 2022-09-14

Accept